# Unraveling the Neuroprotective Effect of Natural Bioactive Compounds Involved in the Modulation of Ischemic Stroke by Network Pharmacology

**DOI:** 10.3390/ph16101376

**Published:** 2023-09-28

**Authors:** Juan Carlos Gomez-Verjan, Emmanuel Alejandro Zepeda-Arzate, José Alberto Santiago-de-la-Cruz, Edgar Antonio Estrella-Parra, Nadia Alejandra Rivero-Segura

**Affiliations:** 1Dirección de Investigación, Instituto Nacional de Geriatría (INGER), Blvd. Adolfo Ruiz Cortines 2767, Mexico City 10200, Mexico; jverjan@inger.gob.mx (J.C.G.-V.); bm.ezepeda@gmail.com (E.A.Z.-A.); alberto.santiago@alumnos.uacm.edu.mx (J.A.S.-d.-l.-C.); 2Laboratorio de Fitoquímica, UBIPRO, FES-Iztacala, Unidad Nacional Autónoma de México, Av. De los Barrios No.1, Los Reyes Iztacala, Tlalnepantla 54090, Mexico; estreparr@iztacala.unam.mx

**Keywords:** natural bioactive compounds, natural products, network pharmacology, neuroprotection, ischemic stroke, flavonoids, terpenoids

## Abstract

Ischemic stroke (IS) is one of the leading causes of mortality worldwide. It is characterized by the partial or total occlusion of arteries that supply blood to the brain, leading to the death of brain cells. In recent years, natural bioactive compounds (NBCs) have shown properties that ameliorate the injury after IS and improve the patient’s outcome, which has proven to be a potential therapeutic strategy due to their neuroprotective effects. Hence, in the present study, we use both systems pharmacology and chemoinformatic analyses to identify which NBCs have the most potential to be used against IS in clinics. Our results identify that flavonoids and terpenoids are the most studied NBCs, and, mainly, salidrosides, ginkgolides A, B, C, and K, cordycepin, curcumin, baicalin, resveratrol, fucose, and cannabidiol, target the main pathological processes occurring in IS. However, the medicinal chemistry properties of such compounds demonstrate that only six fulfill such criteria. However, only cordycepin and salidroside possess properties as leader molecules, suggesting that these compounds may be considered in developing novel drugs against IS.

## 1. Introduction

Ischemic stroke (IS) is one of the leading causes of mortality worldwide [1,2]. It is characterized by the partial or total occlusion of arteries that supply blood to the brain, leading to the massive death of brain cells [3]. Currently, canonical IS management relies on the revascularization and prevention of secondary neuronal injury by combining both thrombolytic (recombinant tissue plasminogen activator—rtPA) and surgical procedures, also known as endovascular therapies [4]. However, these therapies have many drawbacks. For instance, the rtPA administration has a narrow window of time (4.5 h of stroke onset), and a tiny proportion of individuals develop symptomatic hemorrhage, and it is recognized that rtPA has neurotoxic effects [5]. On the other hand, despite the newest mechanical devices to perform surgical procedures (mechanical thrombectomy) that are more effective (restoring the blood supply in almost 90% of the cases), these only focus on restoring blood supply to the brain (stopping the ischemia) but do not prevent the damage caused by the reperfusion [6]. In this sense, current research focuses on neuroprotective and neurorestorative therapies by identifying and developing novel compounds to overcome the current drugs’ drawbacks. In this sense, natural products, a broad term encompassing either primary or secondary metabolites from plants, marine products, and microorganisms, represent a valuable source for developing and obtaining small molecules. In fact, from 1981 to 2019, about 36.3% of the clinically used drugs approved by the FDA (antimicrobial, antiparasitic, and anticancer treatments) were natural products or derivatives [7,8]. Particularly in the field of IS, natural bioactive compounds (NBCs), a subset of natural products that have biological activity [9,10,11], are recognized as a rich source of compounds with beneficial effects against the most common pathological effects induced by such conditions [12]. For instance, many authors report that different NBCs reduce infarct size by targeting oxidative stress. However, several NBCs also decrease neuroinflammation due to their immunomodulatory properties (reducing proinflammatory cytokines), inhibiting microglia activation, preventing excitotoxicity and apoptosis, and improving neuroplasticity and endothelial stiffness [13,14,15,16,17,18]. 

Moreover, network pharmacology has emerged as an exciting field that combines biology, bioinformatics, and network science to analyze the interaction among bioactive compounds, drugs, genes, proteins, or other biological molecules, leading to improved identification of new drug targets, the design of new drugs, and the prediction of their side effects [19,20]. In this sense, the IS research field has also benefited from network pharmacology since several studies aim to identify potential drug targets and molecular mechanisms induced by NBCs against IS [21,22,23]. Nevertheless, such studies focus only on a compound or plant extract. Only one published study summarizes the most commonly used NBCs in traditional Chinese medicine. It identifies the primary molecular targets of such NBCs [24] but does not perform cheminformatic analysis to rationally suggest which NBCs are more suitable to be used in therapeutics. Therefore, in the present article, we aim to identify which NBCs have potential to be used against IS in clinics through both systems pharmacology and chemoinformatic analyses. First, we perform a systematic review of the literature published in the last 12 years regarding NBCs used against IS, and then we retrieve the most relevant data for the network pharmacology analyses; these analyses lead us to identify the most connected NBCs in the network, and finally, such NBCs were subjected to chemoinformatic analyses to identify which NBCs are more suitable for success in the clinics. So, with this study, we provide a guide on trends in research towards the neuroprotective effects of NBCs. 

## 2. Results

### 2.1. Twelve Chemical Classes of NBCs Target the Principal Pathological Processes Elicited by IS, but Only Flavonoids and Terpenoids Are the Most Studied

According to our research in the literature published between 2010 and 2022, there is a lack of studies that report which NBCs are the most studied against IS, compared to the vast number of narrative reviews describing the biological activities and molecular mechanisms elicited by such NBCs used against IS. In this context, our results from Figure 1A demonstrate that, even though twelve chemical classes of NBCs have been tested against IS, only flavonoids (27 studies) and terpenoids (13 studies) are the most studied chemical classes of NBCs. In contrast, the remaining NBCs had fewer studies in the period covered by our search; for instance, polyphenols (10 studies), alkaloids (9 studies), saponin and iridoid glycosides (8 studies each), glycosides (5 studies), phytocannabinoids (4 studies), adenosine analogues and N-linked glycans (3 studies each), anthraquinones (2 studies), and trans-cinnamaldehyde (1 study). Additionally, this result is validated by the results from Figure 1B, since according to the structural network analysis, flavonoids are the most connected nodes in the network (49 edges) in comparison with other NBCs such as terpenoids (23 edges), polyphenols (17 edges), and alkaloids (12 edges), meaning that flavonoids may interact with more targets associated with IS than other compounds. 

Once we had results from the first network, we built another network with NBCs and the pathological processes they target (Figure 1C). According to the results, the main targets of the NBCs against IS are infarct size, microglia activation, oxidative stress, neuroinflammation, neurotransmission, glutamate excitotoxicity, blood-brain barrier (BBB) integrity, autophagy, mitochondrial dysfunction, and apoptosis. Nevertheless, since the chemical classes of these NBCs include more than hundreds of compounds, we aim to identify which natural compounds target each process mentioned above. 

### 2.2. Network of NBCs and the Most Common Pathological Pathways Associated with IS

As seen in the structural network from Figure 2A, there are 34 NBCs (astragaloside, *Panax notoginseng* saponin-TSPN, and ginsenoside Rg1, picroside II, catalpol, and geniposide, cannabidiol, fucose, baicalin, scutellarin, vitexin, apigenin, icariin, quercetin, calycosin, xanthohumol, carthamin yellow, and dihydromyricetin, berberine, ligustrazine, daurisoline, tetrahydropalmatine, neferine, curcumin, resveratrol, asiaticoside and salidroside, andrographolide, ginkgolides (A, B, C, K), borneol, and daidzein, emodin, cordycepin, and cinnamaldehyde) which target the most common pathological pathways associated with IS. According to the structural network from Figure 2B, the most connected nodes are ginkgolides (A, B, C, and K), curcumin, baicalin, fucose, cannabidiol, resveratrol, cordycepin, icariin, and salidroside. Additionally, we built another network showing the most connected NBCs and the molecular targets of such compounds (Figure 2C). As a result, the networks show that each NBC has its own molecular targets but also shares other molecular targets such as IL1-β, Bax, Bcl-2, SOD, TNF-α, Caspase-3, IL-8, IL-6, NF-κB, BDNF, GFAP, and Iba-1, suggesting that the development of novel drugs against IS may be directed towards these targets. Moreover, this network shows that the selected NBCs seem to decrease the expression or protein content of the molecular target since the number of green edges is higher than the number of red edges colored red. 

### 2.3. Chemoinformatic Analysis of the NBCs

Once we identify the most connected NBCs that target the main pathological processes in IS, we perform a chemoinformatic analysis to identify which compounds are more suitable to become therapeutic agents against IS. According to our results (Table 1), cordycepin, curcumin, fucose, ginkgolide A, ginkgolide B, resveratrol, and salidroside meet Lipinski’s rules. However, once we analyze the pharmacokinetic simulated properties, our results suggest that ginkgolides B, C, and K are not suitable for therapeutics since the gastrointestinal absorption is lower in contrast to other NBCs such as cannabidiol, cordycepin, curcumin, fucose, ginkgolide A, resveratrol, and salidroside. In addition, the medicinal chemistry properties suggest that ginkgolides A, B, C, and K are unsuitable for chemical synthesis. 

Interestingly, only cordycepin and salidroside (Figure 3) possess remarkable pharmacokinetic properties and meet the lead-likeness criteria, suggesting they could be considered hits and leaders in developing novel drugs against IS. 

## 3. Discussion

IS represents a global health problem since it is the second cause of mortality and disability worldwide. Hence, current research in the field seeks novel protocols that improve IS diagnosis and treatment to overcome adverse outcomes [25,26]. As a result of this concern, several approaches have been developed; one of them relies on identifying natural products that target the main pathological processes underlying IS [27,28]. However, despite the significant number of original articles or reviews that seek to highlight the relevance of considering the use of natural products against IS, they limit themselves to adding more evidence of the effectiveness of the natural compounds or summarizing the results from other studies without suggesting the next step in the development of novel drugs or their application in clinical practice. Hence, to fill the gaps in the knowledge mentioned above, in the present study, we aim to identify which NBCs are the most studied against IS and then suggest which of them have the potential to formulate and design more effective pharmacological treatments against the pathological processes elicited by IS. 

Accordingly, we perform a network pharmacology analysis based on a systematic review, demonstrating that flavonoids are the most studied compounds against IS; this result may be because flavonoids are enriched in many herbal products and, concomitantly, since their chemical structure (the hydroxy groups) influences their bioavailability and pharmacological activity [29]. For instance, flavonoids are associated with several beneficial biological properties such as anti-inflammatory, anticancer, anti-aging, cardio-protective, neuroprotective, immunomodulatory, antidiabetic, antibacterial, antiparasitic, and antiviral [30,31,32]. On the other hand, the same network identifies terpenoids as the second most studied NBCs against IS. Terpenoids, a class of terpenes with different functional groups and oxidized methyl groups, are the second most widespread NBCs that show antiviral, antibacterial, antimalarial, anti-inflammatory, anticancer, and hypoglycemic activities, and such compounds are found in plants, fungi, and some animals [33,34,35]. Additionally, these results also suggest that studying a plant or marine product rich in flavonoids or terpenoids is more likely to have neuroprotective effects against IS. Nevertheless, it is essential to note that clinical trials in humans are necessary to confirm their effects in real-world scenarios. Further studies may be conducted to develop a library of compounds derived or inspired by flavonoids and terpenoids to treat IS efficiently. 

Since the chemical classes of NBCs include more than hundreds of compounds, we performed another network pharmacology analysis to identify which NBCs target each pathological process. Such a network identified nine compounds (ginkgolides (A, B, C, K), curcumin, baicalin, fucose, cannabidiol, resveratrol, cordycepin, icariin, and salidroside) as the most connected nodes in the network since they target six out of eleven of the most common pathological processes induced by IS (infarct size, neuroinflammation, apoptosis, oxidative stress, neuroplasticity, and microglia activation). Interestingly, many of these compounds are found in herbal medicines. For instance, ginkgolides (A, B, C, and K), a group of terpene trilactones, are isolated from the *Ginkgo biloba* tree and have shown a beneficial role in health and wellbeing since it is well described that ginkgo extract induces antiparasitic, antifungal, antibacterial, and antiviral activities [36]. Interestingly, it has been reported that the bioavailability of ginkgolides is low; however, the preparation method of ginkgol extracts influences the bioavailability [37]. 

On the other hand, curcumin (1,7-bis-(4-hydroxy-3-methoxyphenyl)-hepta-1,6-diene-3,5-dione) isolated from *Curcuma longa* belongs to the polyphenol group [38,39,40]. It has been reported that curcumin has enormous therapeutic potential since it exhibits anti-inflammatory, antioxidant, antiviral, proapoptotic, chemo-preventive, chemotherapeutic, antinociceptive, antiproliferative, antiparasitic, and antimalarial effects [41]. However, poor absorption, rapid metabolism, and systemic elimination, leading to poor bioavailability, are curcumin’s main limitations in reaching therapeutics [42]. Moreover, baicalin (5,6-dihydroxy-7-O-glucuronide flavone) is the main active compound of the medicinal plants *Scutellaria baicalensis* and *Oroxylum indicum* [43]; such NBC is a flavonoid that has antibacterial, antiviral, anticancer, anticonvulsant, antioxidant, hepatoprotective, and neuroprotective effects in experimental models; however, its use is limited in clinical trials due to poor bioavailability [44,45]. Conversely, fucose is a monosaccharide that induces biological activities such as anticancer, anti-allergic, anti-coagulant, and anti-aging. Nevertheless, despite the great potential that fucose represents for the food, cosmetics, and pharmaceutical industries, its main limitation is the complexity and high cost of performing its chemical synthesis [46,47]. Likewise, cannabidiol represents the most abundant and therapeutically relevant component in *Cannabis sativa*, *Cannabis indica*, and *Cannabis ruderalis*. According to the literature, cannabidiol induces neuroprotective, antiepileptic, anxiolytic, antipsychotic, and anti-inflammatory effects. However, this NBC has bioavailability and solubility issues, and current studies have focused on developing a novel series of CBD analogues to overcome such drawbacks [48,49]. Similarly, resveratrol (3,5,4’-Trihydroxystilbene) is a polyphenol with a stilbene structure isolated from *Veratrum grandiflorum*. Resveratrol has antioxidant, anti-inflammatory, cardioprotective, antimicrobial, and anti-tumorigenic properties [50]. However, resveratrol also shows poor solubility in water, leading to low oral bioavailability; also, resveratrol has a shorter half-life when administered intravenously [51,52]. Finally, icariin is an isoprenoid flavonoid that represents the active compound of *Epimedii herba* [53,54]; this NBC induces anti-inflammatory, antioxidant, antidepressant, and aphrodisiac effects [55,56]; however, a Phase I trial demonstrated that icariin has low bioavailability of the oral formulation [57]. So, this growing body of evidence suggests that the main limitation of the NBCs we identified by our network analysis is their low bioavailability and the complexity and high cost of performing their chemical synthesis. Our chemoinformatic analysis strengthens such results, as discussed in the following paragraphs. 

Since several drugs failed in clinical trials due to their pharmacokinetic properties, we performed a chemoinformatic analysis of the medicinal chemistry properties of the nine NBCs described above so we may identify which NBC may have more potential to be used to alleviate the most common pathological processes in IS. Such analysis shows that only cordycepin, curcumin, fucose, ginkgolide A, ginkgolide B, resveratrol, and salidroside fulfill Lipinski’s rule of five. However, only cordycepin and salidroside fulfill lead-likeness rules since neither contains any toxicoinformatics properties. Interestingly, neither of the NBCs are flavonoids, suggesting that despite flavonoids being the more studied NBC against IS, salidroside (phenolic glycoside) and cordycepin (adenosine analogue) may be easier to incorporate as therapeutic drugs against IS. Moreover, these results suggest that both NBCs could be considered in developing novel strategies to develop drugs against IS or to search for natural products with a high content of these compounds. In this sense, cordycepin is an alkaloid and the primary bioactive molecule derived from *Cordyceps,* a well-known fungus used in Chinese traditional medicine for the last 300 years to treat fatigue, sickness, kidney disease, and low sex drive [58]. Various IS-related activities have been described for this NBC, such as anti-inflammatory, inhibition of platelet aggregation, and immunomodulatory effects [58,59]. On the other hand, salidroside is the main bioactive component in *Rhodiola* species; it is a phenylpropanoid glycoside that has also been associated with IS-related pathways, such as suppressing oxidative stress, inflammation, and enhancing cell survival [60]. 

Finally, pure compounds alone may be helpful; however, a cocktail of NBCs could represent a viable alternative in searching for IS treatments with a novel increase in phytomedicine. In this context, it is pretty interesting that there is a renewed interest in developing this type of product due to the success of NeuroAID [61], recently approved in China, Singapur, South Korea, Indonesia, and Vietnam to be used on IS and composed of nine herbal components. Thus, expectations are great for these types of products. 

## 4. Materials and Methods

### 4.1. Data Collection

Following the PRISMA statement [62], we systematically reviewed the literature published between January 2010 and January 2022 (Figure 4). MESH terms, including ‘*cerebral ischemia*’, ‘*natural medicine*’, ‘*stroke*’, ‘*neuroprotection mechanism*’, ‘*natural products*’, ‘*therapeutic applications*’*,* and ‘*natural extracts*’, were submitted to the Pubmed database (https://pubmed.ncbi.nlm.nih.gov/, accessed on 30 March to 30 April 2023). Two independent reviewers curated the selected articles according to the following criteria: 

Inclusion criteria: Studies published between January 2010 and January 2022 (12 years of research window), original research articles, studies focused on neuroprotection induced by NBCs in IS models, and studies performed on in vitro or in vivo models using NBCs against IS.

Exclusion criteria: Studies that have not met the inclusion criteria, studies published in languages excluding English, articles that were narrative reviews, intervention studies, letters to editors or non-original articles, preprints or studies with incomplete datasets, studies without available data, studies without controls, studies focused on neuroprotection for neurodegenerative diseases (e.g., Huntington’s disease, Alzheimer’s disease, Parkinson’s disease, among others), and articles not available in the PubMed database. 

We retrieved all the relevant information from each selected study, as depicted in Appendix A.

### 4.2. Network Pharmacology Analysis

Pharmacological networks were built using Cytoscape (v 3.9.1) [63,64]. Networks were built with the data retrieved from the selected articles (Appendix A). Additionally, we identify the most connected nodes in the networks using the Cytohubba plugin available on the Cytoscape webpage [65].

### 4.3. Cheminformatic Analyses

Molecular descriptors (1-3D) were calculated for each NBC using three software and web tools: DruLiTo v.1.0 (Nagar, India) is open-source software that can calculate different molecular properties and screen molecules based on pharmacological property rules such as the Lipinski rule, MDDR rule, Veber rule, Ghose filter, BBB rule, CMC-50 rule, and quantitative estimation of pharmacological properties (QED) [66]; DataWarrior v.5.5.0 (Allschwil, Switzerland) [67], an open-source program for visualization and data analysis with chemical intelligence following the methodology of [67]; SwissADME (http://www.swissadme.ch/, Lausanne, Switzerland), a website that allows calculating physicochemical descriptors as well as predicting ADME parameters, pharmacokinetic properties, pharmacological nature, and medical chemical compatibility of single or multiple small molecules to support drug discovery. Such analysis was based on Lipinski’s rules: molecular weight (MW) equal to or less than 500 Da, octanol-water partition coefficient equal to or less than 5 (cLogP), the total number of hydrogen acceptors equal to or less than 10 (HBA), and the total number of hydrogen donors equal to or less than 5 (HBD) [68,69,70] and pharmacokinetics [71]. 

## 5. Conclusions

In the present article, we resume most of the studies performed on natural products and IS through 12 years of research to guide trends in research towards the neuroprotective effects of NBCs against IS. In this sense, we found that flavonoids are the most studied NBCs tested against IS to date (2023). We also identify that of the nine NBCs that target the most common pathological process in IS, only cordycepin (adenosine analogue) and salidroside (phenolic glycoside) may be considered leaders to be used in developing chemical libraries and novel molecules against such a pathology. Finally, our study highlights the importance of searching for plant extracts that contain these compounds, which could represent the source for phytomedicine development. 

## Figures and Tables

**Figure 1 pharmaceuticals-16-01376-f001:**
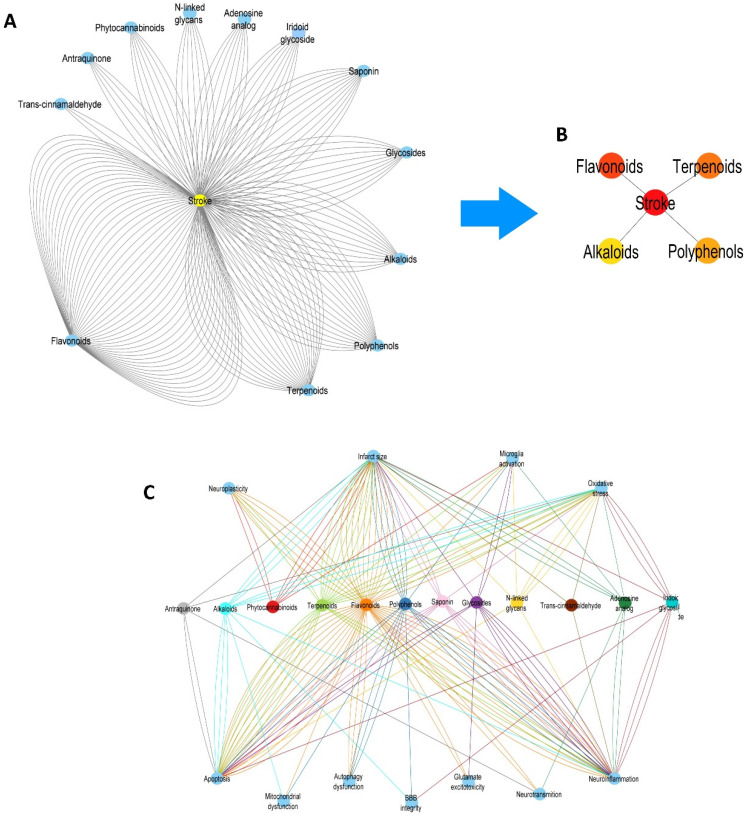
Chemical classes of the NBCs studied against IS. (**A**) Blue nodes represent the NBCs (flavonoids, terpenoids, polyphenols, alkaloids, glycosides, saponin, iridoid glycosides, phytocannabinoids, adenosine analogues, N-linked glycans, anthraquinones, and trans-cinnamaldehyde). Edges (black) represent a single pathological target for each node. Number of nodes: 15, number of edges: 166, network diameter: 1. (**B**) The structural network represents the top five of the most connected nodes in the network. The most connected nodes in the network are flavonoids and terpenoids. The most connected nodes are colored red, and the less connected nodes are colored yellow. (**C**) The structural network shows the chemical classes (central nodes) considered in this study (saponin (pink), iridoid glycoside (aqua), phytocannabinoids (red), N-linked glycans (yellow), flavonoids (orange), alkaloids (light blue), polyphenols (blue), terpenoids (light green), anthraquinone (grey), adenosine analogues (green), and trans-cinnamaldehyde (brown)). Edges connect these nodes (blue) to the main pathological processes that underlie IS. Each edge represents an individual NBC derived from their chemical class. The edges are colored according to the chemical class of each compound. Number of nodes: 23, number of edges: 163, network diameter: 1.

**Figure 2 pharmaceuticals-16-01376-f002:**
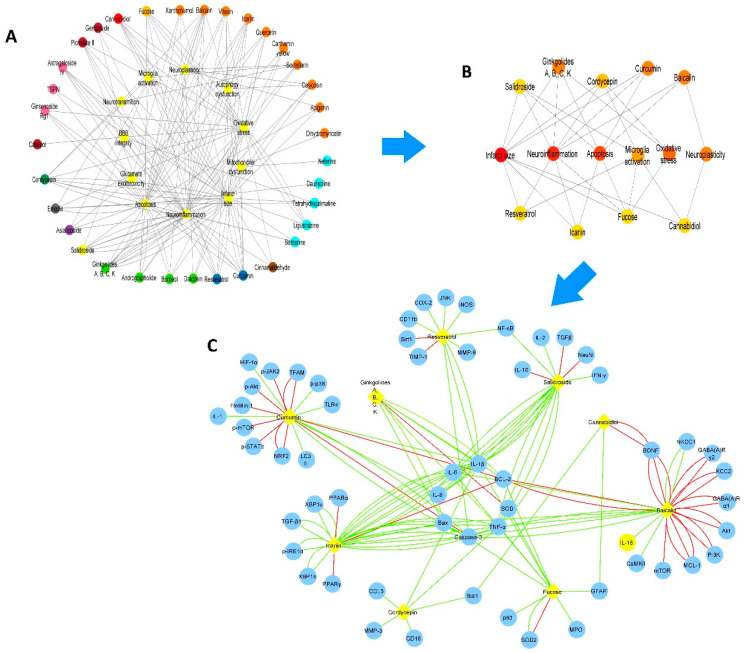
Nine NBCs are required to target the most common pathological processes that underlie IS. (**A**) The network shows the pathological processes that each NBC targets in IS. Number of nodes: 45; number of edges: 163; and network diameter: 1. (**B**) The structural network represents the top 15 of the most connected nodes in the network. Nine NBCs (ginkgolides (A, B, C, K) (terpenoid), curcumin (polyphenol), baicalin (flavonoid), fucose (N-linked glycans), cannabidiol (phytocannabinoids), resveratrol (polyphenol), cordycepin (adenosine analogue), icariin (flavonoid), and salidroside (glycoside) target the main pathological processes that underlie IS (infarct size, neuroinflammation, apoptosis, oxidative stress, neuroplasticity, and microglial activation). The most connected nodes are colored red, and the less connected nodes are colored yellow. (**C**) The network shows the molecular targets of the pathological pathways in IS (blue nodes) connected to the most connected compounds (yellow nodes). The edges represent the effect of the NBCs on each target; green lines represent decreased expression; and red lines represent increased expression. Each edge represents an independent report. In the network’s center are the targets shared among the NBCs. Number of nodes: 64; number of edges: 141; and network diameter: 3.

**Figure 3 pharmaceuticals-16-01376-f003:**
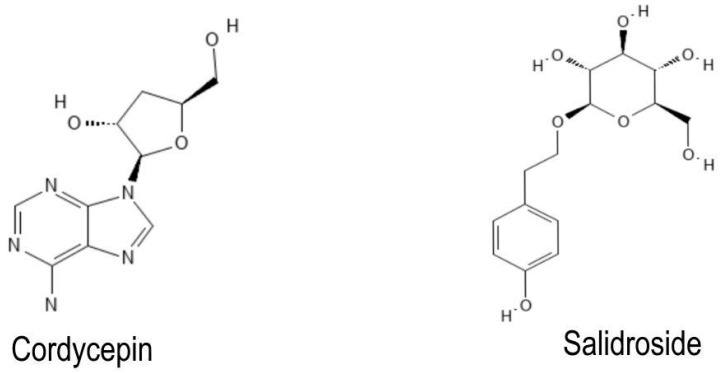
The chemical structure of both NBCs met the lead-likeness criteria.

**Figure 4 pharmaceuticals-16-01376-f004:**
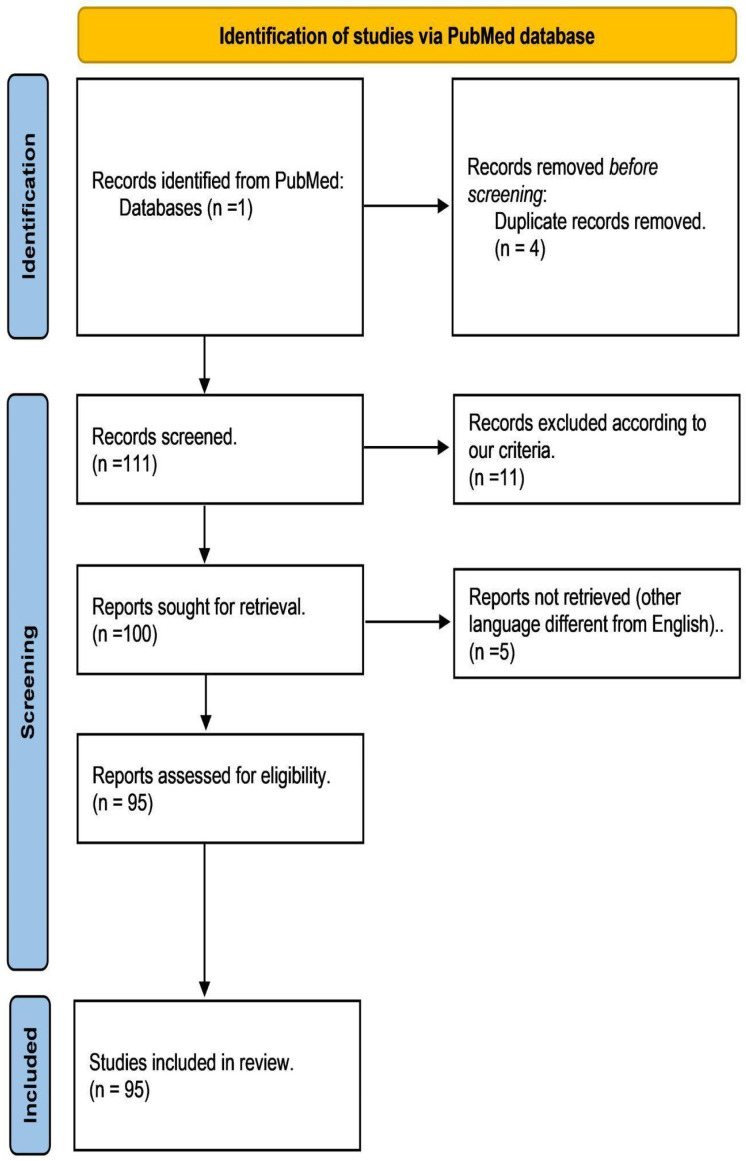
PRISMA flow diagram of the article selection for the systematic review of NBCs against IS. According to our selection criteria for this study, we selected 95 articles, from which we retrieved the following information: chemical class, chemical compound name, experimental model (in vivo or in vitro), molecular target, treatment (doses), biological activity, and reference (Appendix A). (http://prisma-statement.org/prismastatement/FlowDiagram.aspx, accessed on 1 January 2022) [62].

**Table 1 pharmaceuticals-16-01376-t001:** Molecular descriptors of the most connected natural compounds against IS.

		Baicalin	Cannabidiol	Cordycepin	Curcumin	Fucose	Ginkgolide A	Ginkgolide B	Ginkgolide C	Ginkgolide K	Icariin	Resveratrol	Salidroside
Physicochemical Properties	Log P	1.051	6.158	−2.195	1.945	−0.994	1.176	0.146	−1.095	0.914	1.092	2.048	−1.071
	Log S	−2.724	−4.493	−2.724	−3.622	−0.256	−2.526	−2.127	−1.728	−2.277	−4.133	−2.864	−1.016
	TPSA	183.21	40.46	116.03	93.06	90.15	128.59	148.82	169.05	128.59	234.29	60.69	119.61
	MW	446.08	314.46	251.1	368.38	164.16	408.4	424.4	440.13	406.38	676.24	228.24	300.3
	nRB	4	6	2	8	0	1	1	1	1	9	2	5
	HBD	6	2	3	2	4	2	3	4	2	8	3	5
	HBA	11	2	8	6	5	9	10	11	9	15	3	7
Pharmacokinetic Properties	GI absorption	Low	High	High	High	High	High	Low	Low	Low	Low	High	High
	BBB permeable	No	Yes	No	No	No	No	No	No	No	No	Yes	No
	P-gp substrate	Yes	No	No	No	Yes	Yes	Yes	Yes	Yes	Yes	No	No
	CYP1A2 inhibitor	No	No	No	No	No	No	No	No	No	No	Yes	No
	CYP2C19 inhibitor	No	Yes	No	No	No	No	No	No	No	No	No	No
	CYP2C9 inhibitor	No	Yes	No	Yes	No	No	No	No	No	No	Yes	No
	CYP2D6 Inhibitor	No	Yes	No	No	No	No	No	No	No	No	No	No
	CYP3A4 inhibitor	No	Yes	No	Yes	No	No	No	No	No	No	Yes	No
	Log Kp (Skin permeation)	−8.23	−3.59	−8.27	−6.28	−8.79	−8.37	−9.16	−9.95	−9.95	−9.25	−5.47	−8.88
Medicinal Chemistry Properties	Lipinski violations	2	1	0	0	0	0	0	1	1	3	0	0
	Ghose violations	0	1	1	0	2	0	1	1	1	3	0	1
	Veber violations	1	0	0	0	0	0	1	1	1	1	0	0
	Bioavailability Score	0.11	0.55	0.55	0.55	0.55	0.55	0.55	0.55	0.55	0.17	0.55	0.55
	Lead-likeness violations	1	1	0	2	1	1	1	1	1	2	1	0
	Synthetic accessibility	5.09	4.05	3.67	2.97	4.05	6.28	6.38	6.48	6.48	7.24	2.02	4.26
Toxicoinformatic Properties	Mutagenic	none	none	none	none	none	none	none	none	none	high	high	none
	Tumorigenic	none	none	none	none	none	none	none	none	none	none	none	none
	Irritant	none	none	none	none	none	none	none	none	none	none	none	none
	Reproductive effects	none	none	none	none	none	none	none	none	none	high	high	none

## Data Availability

Data is contained within the article.

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
