# Peer review of "Unraveling the Neuroprotective Effect of Natural Bioactive Compounds Involved in the Modulation of Ischemic Stroke by Network Pharmacology"

_pharmaceuticals, 2023, doi:10.3390/ph16101376_

Round 1
Reviewer 1 Report
The manuscript entitled 'Unravelling neuroprotective effect of natural products involved in the modulation of ischemic stroke by network pharmacology ' needs a major revision.
Main notes:
1. The title of the paper 'Unravelling neuroprotective effect of natural products involved in the modulation of ischemic stroke by network pharmacology' could be little changed – for instance, into 'Unravelling neuroprotective effect of natural products compounds involved in the modulation of ischemic stroke by network pharmacology'.
2. A minor check is required for the English language and style through the text. For instance, in the Abstract: line 23 – six fulfill.
3. The list of keywords for searching for information should be expanded by the authors, for instance using 'flavonoids ischemic stroke' https://pubmed.ncbi.nlm.nih.gov/?term=flavonoids+ischemic+stroke&sort=date&size=20 (403 results) or 'terpenoids ischemic stroke' (765 results in PubMed). Generally, authors elaborated too little (only 8 sources!) of new scientific articles published in the last 3 years. Thus, the list of references should be complemented and renewed significantly! There is a lot of such data in the PubMed database.
4. I believe that the list of used sources (references) should be expanded at least twice.
5. Line 55 - The abbreviation 'ROS' occurs only once in the text, so it should be removed.
6. Lines 254-257 - The authors claim in the first half of the sentence that flavonoids are the most active components in relation to IS, but in the second half they cite cordycepin and salidroside as the most active compounds. However, they are not flavonoids, so that needs some explaining: 'In this sense, we found that flavonoids are the most active compounds tested against IS at the date and that Cordycepin and Salidroside may be considered leaders to be used in developing chemical libraries and novel molecules against such a pathology'
7. For a better visualization and generalization of the analized literary sources, the authors should create a Table in which data from articles with experimental (indicating the model used) and clinical studies on each of the described natural components regarding the pharmacological activity in relation to IS will be demonstrated.
8. The title of this subsection is inappropriate because the text describes more than flavonoids and terpenoids '2.1. Flavonoids and Terpenoids are the most active compounds against IS'.
9. Lines 191-192 - In this sentence, for some reason, the names of the compounds are capitalized, although in most places in the text they are written in lowercase 'In this sense, we found that only Cordycepin, Curcumin, Fucose, Ginkgolide A, Ginkgolide B, Resveratrol and Salidroside…. ''
10. The italic type should be used everywhere for writing Latin names of species and for terms in vitro, in vivo for instance - Lines 222, 223, etc.
A minor check is required for the English language and style through the text.
Reviewer 2 Report
This is an interesting manuscript on potential neuroprotectors against ischemic stroke. However, there are several issues that authors should resolve before consideration for article publication.
First, authors should consider rephrasing the title to be more concise. Also, based on the provided manuscript content, instead of natural products, it is more appropriate to use the term “bioactive compounds” or similar.
Please clarify the aim(s) of the study in the Abstract and at the end of the Introduction.
In the first four paragraphs of the Discussion, there are no references.
Please uniform presenting names of phytochemicals/bioactive compounds throughout the whole manuscript—no need to capitalize the first letter.
The conclusion needs to be based on obtained data. Are flavonoids the most studied compounds or more active compounds? Please provide data to support this conclusion.
Discussion should be supplemented with more data on identified bioactive compounds. Are there limitations or risks for their potential clinical applications?
Where are extracted data from included studies?
Some corrections are needed.
Round 2
Reviewer 1 Report
The manuscript was substantially improved by the authors.
In the Conclusion section (lines 337-338) it could be added the word 'phenolic' in the parenthesis 'and salidroside (phenolic glycoside) may be...'
Author Response
Response to Reviewers
Reviewer 1
Q1. The manuscript was substantially improved by the authors. In the Conclusion section (lines 337-338) it could be added the word 'phenolic' in the parenthesis 'and salidroside (phenolic glycoside) may be...'
Answer: Thank you for your valuable observation. Following, the word "phenolic" has been added. Please refer to the main manuscript for more information.
Reviewer 2 Report
The authors have addressed most of the earlier concerns and improved the manuscript.
The manuscript would benefit from professional English editing.
Author Response
Response to Reviewers
Reviewer 2
Q1. The authors have addressed most of the earlier concerns and improved the manuscript.
The manuscript would benefit from professional English editing
Answer. Thank you for your valuable commentary. Following your suggestion, our manuscript has been edited by a native English speaker. Please refer to the main manuscript for more information.